# A Splicing Variant in RDH8 Is Associated with Autosomal Recessive Stargardt Macular Dystrophy

**DOI:** 10.3390/genes14081659

**Published:** 2023-08-21

**Authors:** Stefania Zampatti, Cristina Peconi, Giulia Calvino, Rosangela Ferese, Stefano Gambardella, Raffaella Cascella, Jacopo Sebastiani, Benedetto Falsini, Andrea Cusumano, Emiliano Giardina

**Affiliations:** 1Genomic Medicine Laboratory UILDM, IRCCS Santa Lucia Foundation, 00179 Rome, Italy; s.zampatti@hsantalucia.it (S.Z.);; 2Neuromed IRCSS, 86077 Pozzilli, Italy; 3Department of Biomolecular Sciences, University of Urbino “Carlo Bo”, 61029 Urbino, Italy; 4Department of Biomedical Sciences, Catholic University Our Lady of Good Counsel, 1000 Tirana, Albania; 5Macula & Genoma Foundation, 00133 Rome, Italy; jacopo.seba@gmail.com (J.S.);; 6Department of Ophthalmology, Policlinico A. Gemelli, IRCCS/Catholic University, 00133 Rome, Italy; 7Macula & Genoma Foundation USA, New York, NY 10017, USA; 7Department of Ophthalmology, Tor Vergata University, 00133 Rome, Italy; 8Department of Biomedicine and Prevention, Tor Vergata University, 00133 Rome, Italy

**Keywords:** Stargardt disease, macular dystrophy, RDH8, all-trans-retinol dehydrogenase, retinal disease

## Abstract

Stargardt macular dystrophy is a genetic disorder, but in many cases, the causative gene remains unrevealed. Through a combined approach (whole-exome sequencing and phenotype/family-driven filtering algorithm) and a multilevel validation (international database searching, prediction scores calculation, splicing analysis assay, segregation analyses), a biallelic mutation in the *RDH8* gene was identified to be responsible for Stargardt macular dystrophy in a consanguineous Italian family. This paper is a report on the first family in which a biallelic deleterious mutation in *RDH8* is detected. The disease phenotype is consistent with the expected phenotype hypothesized in previous studies on murine models. The application of the combined approach to genetic data and the multilevel validation allowed the identification of a splicing mutation in a gene that has never been reported before in human disorders.

## 1. Introduction

The first description of Stargardt macular dystrophy was published by Karl Bruno Stargardt in 1909 [1]. Macular dystrophy was defined as a genetic disorder that compromised cone, retinal pigmented epithelium (RPE), and choroid functions. The “fundus flavimaculatus” was successively described by Franceschetti [2]. During the last century, the clinical characterization of *ABCA4*-related Stargardt macular dystrophy revealed a high phenotypic variability [3]. Furthermore, several Stargardt-like macular dystrophies have been described, in which other genes were recognized (*ELOVL4*, *PROM1*, and *PRPH2*). Moreover, macular disorders associated with mutations in other genes (e.g., *CRX*, *MT-TL1*, *RDH12*, and *RPGR*) could be challenging to distinguish from the stages of *ABCA4*-macular dystrophy. To date, the availability of NGS as a support tool in the genetic characterization of patients with ocular disorders makes it possible to identify mutations, even in genes not traditionally associated with the disease.

Despite a huge clinical heterogeneity, patients in which the macular disorder shows a relationship with biallelic mutations in the *ABCA4* gene show three typical signs: progressive atrophy of the cellular layers in the central macula beginning from the RPE, fundus flecks derived from the accumulation of RPE lipofuscin in the retina, and peripapillary sparing (the tissue adjacent to the optic nerve is generally spared of disease changes). The latter is the most unusual sign, and it can be recognized also in non-*ABCA4* retinopathies. From a clinical point of view, the only sign that can be recognized in all the affected patients is maculopathy: the disease always begins in the central macula. However, the age at onset, disease progression, and family history can vary considerably among patients. Monogenic maculopathies comprise over 30 different causative genes [3]. Furthermore, there are some environmental factors that can cause maculopathy (i.e., hydroxychloroquine toxicity, light damage, etc.) [4,5]. Despite the application of advanced sequencing technologies, in about 1 patient out of 10, no biallelic mutations in *ABCA4* have been identified [3]. Some of these cases recognized other causative genes and have been comprehensively classified as phenocopies of *ABCA4*-associated maculopathy. An accurate clinical and familial evaluation sometimes can address the extensive genetic analysis. In this scenario, the evaluation of the family history and the definition of the expected model of inheritance and penetrance may help in prioritizing gene analysis. Furthermore, an accurate evaluation of the whole clinical presentation may help in identifying some pathognomonic non-*ABCA4* associated features (for example angioid streaks in *ABCC6* patients, vitelliform lesions in *BEST1*, or systemic features for genes involved in syndromic disorders).

Despite the availability of a highly advanced platform and technologies for DNA sequencing, some phenocopies of *ABCA4* maculopathy remain unsolved. Some of these cases are due to the complexity of *ABCA4*, which recognizes mutations in regulatory regions, deep-intronic variants, and very rare structural variants [3]. Moreover, in many cases, other responsible genes can be identified, leaving *ABCA4* the most frequent gene in a very long list of potentially involved genes.

In this manuscript, we will describe a whole-exome evaluation of a patient without biallelic mutations in *ABCA4* and genes associated with similar phenotypes. In this family, we recognized a biallelic mutation in *RDH8* [OMIM *608575], evaluated as responsible for a Stargardt-like phenotype.

## 2. Materials and Methods

### 2.1. Patients

The proposita (II:2), a 62-year-old woman, came to our attention with a diagnosis of Stargardt disease. She presented with early signs of maculopathy at 40 years of age. She was evaluated at the age of 62. After ophthalmological evaluation, genetic counselling was performed with the proposita (II:2) and written informed consent and a buccal swab sample were collected. A first post-test genetic counselling session was performed to comment on the results of the screening analysis (negative for mutations in genes mainly associated with retinal disorders, listed in Appendix A). In order to expand the genetic evaluation, a blood sample from the affected sister (II:4) of the proposita was required. Genetic counselling was performed with the affected sister (II:4) to explain the nature and possible consequences of the genetic analysis, and written informed consent was obtained.

Subsequently, to confirm the genotypes detected in II:2 and II:4, segregation analyses were planned for their sons. Genetic counselling was performed with all the sons of II:2 and II:4, and all of them authorized the genetic analysis. A peripheral blood sample was collected for III:1, III:2, III:3, and III:4 (Figure 1). The ophthalmological evaluations performed on III:1, III:2, III:3, and III:4 revealed no retinal abnormalities.

### 2.2. Whole-Exome Sequencing and Bioinformatics Analysis

Genomic DNA was isolated from a buccal swab and blood samples using the MagPurix Blood DNA Extraction Kit and MagPurix Automatic Extraction System (Resnova, Rome, Italy) according to the manufacturer’s instructions. In particular, genomic DNA was extracted from a buccal swab sample from II:2 and a blood peripheral sample from II:2, II:4, III:1, III:2, III:3, and III:4 (Figure 1). The concentration and quality of the extracted DNA were checked using a DeNovix Spectrophotometer (Resnova, Rome, Italy). NGS analysis was performed using an Illumina NextSeq 550 sequencer (Illumina, San Diego, CA, USA) (overall coverage 82.6%). The quality and coverage of the variants were verified using the Integrative Genomics Viewer (IGV). The variant annotation was performed with both INTERVAR/ANNOVAR [6,7] and the BaseSpace Variant Interpreter (Illumina, San Diego, CA, USA). The variant prioritization was first performed by applying a virtual gene panel. Subsequently, a whole-exome analysis was performed on all the potentially pathogenic variants according to the BaseSpace Variant Interpreter or BaseSpace Variant Interpreter.

### 2.3. Variant Validation via Sanger Sequencing

The coding *RDH8* (NM_015725.4) sequence flanking c.262+1G>A mutation was amplified by PCR and direct sequencing. The PCR assay required the following primers: forward primer (5′-CATGCCATAGCCTCAGCACCT-3′) and reverse primer (5′-GAGCTGGTGCTCAGACCTTG-3′). The PCR assay was performed in 25 μL containing 50 ng of genomic DNA, 2 mmol/L of MgCl2, 25 pmol/L of primers, and 1 unit of Taq polymerase (AmpliTaq Gold™ DNA Polymerase, Applied Biosystems, Waltham, MA, USA). The samples were amplified in a 0.2 mL MicroAmp™ 8-Tube Strip (Applied Biosystems, Waltham, MA, USA) in a Veriti™ 96-well Thermal Cycler at a 100% ramping rate. The standard thermal cycling parameters were as follows: enzyme activation at 94 °C for 10 min; 35 cycles of denaturation at 94 °C for 1 min, annealing at 67 °C for 1 min, and extension at 72 °C for 1 min, followed by a final extension step at 72 °C for 7 min.

### 2.4. Effect Evaluation of Variant

The detected variant c.262+1G>A in the *RDH8* gene was evaluated using bioinformatic prediction software and international databases. The variant was not previously reported in the Human Gene Mutation Database (HGMD, www.hgmd.cf.ac.uk, last accession 21 August 2023), ClinVar, or the Leiden Open Variation Database (LOVD) [8]. Combined Annotation Dependent Depletion (CADD) [9], Database Splicing Consensus Single Nucleotide Variant (dbscSNV) [10], FATHMM-MKL [11], DANN [12], and MutationTaster [13] were used to evaluate the potential splicing effect of the variant. Pathological classification was performed according to the American College of Medical Genetics (ACMG) criteria [14,15].

The population frequencies were evaluated according to gnomAD Exome, gnomAD Genome, BRAVO, and All of Us databases. The GnomAD Exome and Genome coverage were checked to evaluate the reliability of the population data.

The potential splicing effect of the *RDH8* variant c.262+1G>A was investigated using the pSPL3 minigene vector (exon trapping system, Gibco, BRL, Carlsbad, CA, USA).

The minigene constructs containing either the wild-type or variant sequence were transfected into HEK293 cells by Lipofectamine 2000 (Invitrogen Corporation, Carlsbad, CA). Forty-eight hours later, the total cellular RNA was isolated using the acidic guanidine phenol-chloroform method. First-strand cDNA was synthesized by SuperScript^®^ VILOTM (Thermo Fisher Scientific, Waltham, MA, USA). RT-PCR was performed using the vector exonic primers SD6 (forward) and SA2 (reverse) according to the manufacturer’s instructions. The final PCR products obtained from transfection with the wild-type and variant plasmids were analyzed by DNA sequencing.

## 3. Results

### 3.1. Clinical Features

The proposita was the first-born of healthy consanguineous parents (second cousins) (Figure 1). She presented early signs of maculopathy at 40 years of age. She was evaluated at the age of 62. The best-corrected visual acuity was 20/25 in both eyes with myopic refraction (−3.50 sph −1 cyl diopters). The intraocular pressure was normal in both eyes (16 mm Hg), but there was a history of ocular hypertension under treatment with prostaglandins. Fundus imaging showed perifoveal and peripheral flecks as well subretinal deposits in the foveal region with thinning of the outer nuclear layer (Figure 2 and Figure 3). The spectral domain optical coherence tomography (OCT) and fundus autofluorescence are shown in Figure 2 and Figure 3, respectively. Figure 4 shows the rod–cone and cone electroretinograms (ERGs) recorded in both eyes of the patient. The ERGs were of normal amplitude in both eyes.

### 3.2. Bioinformatics and Molecular Analyses

Whole-exome sequencing detected 259,941 variants in the proposita’s DNA, including 28 single nucleotide variations (SNV) in the mitochondrial DNA, 601 indels, and 24,463 SNVs in the coding genomic sequence. The first analysis included an evaluation of all the variants detected in a virtual panel of 83 genes (Appendix A). This analysis retrieved no potentially deleterious variants in the selected genes.

Therefore, an extensive analysis of whole-exome variants was performed. The SNVs and indel variants were filtered for coverage (mean coverage > 20x), type (missense, nonsense, frameshift, and splice site mutations), allele frequency (<0.05 according to gnomAD), and potential deleteriousness (according to the BaseSpace Variant Interpreter or InterVar). Fifty-five selected variants were then evaluated to verify a match with the phenotype: 8 variants (4 SNVs and 4 indels) were classified as potentially benign; 11 variants (9 SNVs and 2 indels) in genes with a known OMIM phenotype (heterozygous in autosomal recessive genes) not matching with the ocular phenotype of the case; 9 variants (3 SNVs and 6 indels) in genes without an OMIM record; 20 variants (7 SNVs and 13 indels) in OMIM genes without a known OMIM phenotype. These genes are mainly expressed in tissues other than the retina; 3 indel variants in genes without a known OMIM phenotype. These genes are expressed in the retina and other tissues. These indel variants are classified as variants of unknown significance (VUS), but their genotype (heterozygous variants) was inconsistent with the expected model of inheritance (autosomal recessive); 1 substitution in the *RDH8* gene (NM_015725.2: c.262+1G>A), mainly expressed in the retina and adipocytes, classified as a variant of unknown significance (VUS), with a genotype (homozygous variant) consistent with the expected model of inheritance (autosomal recessive); 3 indel low-quality variants classified as sequencing artifacts. As reported, according to the OMIM database (Online Catalog of Human Genes and Genetic Disorders), no deleterious variants have been reported in genes with a known ocular OMIM phenotype. All the potentially involved variants were evaluated to verify the localization of the gene products and the consistency with the expected model of inheritance.

This analysis allows for the selection of a potentially deleterious variant (NM_015725.2: c.262+1G>A; chr19:10127892:G:A) in the *RDH8* gene. To date, no human phenotypes have been associated with *RDH8* mutations. The localization of the RDH8 enzyme and the broad literature regarding the role of RDH8 in experimental animals supported its potential effect on our family [16,17].

Sanger sequencing was performed on all the affected patients (II:2 and II:4) and their sons and daughters (III:1, III:2, III:3, and III:4) to confirm the expected genotypes. Direct sequencing of the sequence franking c.262+1G>A variant in the *RDH8* gene confirmed homozygosity in the II:2 and II:4 samples. As expected, each son was heterozygous for the c.262+1G>A variant.

Sequencing analysis allowed the confirmation of the co-segregation of the genotype with the phenotype (each affected patient was homozygous for the c.262+1G>A variant in *RDH8*). According to the American College of Medical Genetics and Genomics (ACMG) guidelines [14,15], the c.262+1G>A variant was then classified as pathogenic.

### 3.3. Effect Evaluation of Variant

The c.262+1G>A variant in the *RDH8* gene was not previously reported in the Human Gene Mutation Database (HGMD, www.hgmd.cf.ac.uk (last accession 21 August 2023), ClinVar, or the Leiden Open Variation Database (LOVD) [8]. The population frequencies in the gnomAD dataset reported an allele frequency of 0.00000432 (1 heterozygous sample in the European non-Finnish population in over 100,000 samples). The population frequencies in the BRAVO and All of Us databases were consistent with gnomAD: 0.000004322 and 0.000002, respectively. The GnomAD Exome and Genome reported a mean coverage of 24.8 and 31.9, respectively.

The prediction scores reported a potential pathogenic effect for Combined Annotation Dependent Depletion (CADD) (ADA score: 34) [9], Database Splicing Consensus Single Nucleotide Variant (dbscSNV) (score 0.9999) [10], and FATHMM-MKL (score: 0.9953) [11]. DANN [12] and MutationTaster [13] reported an uncertain effect for this variant (0.9953 and 1.1, respectively).

In order to verify the effects of this genetic variant on the splicing mechanisms, a splicing analysis assay was carried out. The minigene assay is shown in Figure 5. PCR of cDNA without a mutation produces an amplicon of 423 bp related to a wild-type genotype (159 bp of normal splicing of exon 2 + 264 bp of pSPL3 exon), while PCR of cDNA with a mutation produces an amplicon of 492 bp, evidencing abnormal splicing, causing the partial retention of intron 2 (159 bp (normal splicing of exon 2) + 69 bp (intron 2 10.127.891–10.127.960) + 264 bp of pSPL3 exon)). This retention results in an insertion of nine amino acids (GETSQPLIH) and a premature stop codon (Figure 5).

The variant has been submitted to ClinVar and is under evaluation (SUB13728545).

## 4. Discussion

Stargardt disease is the most common type of inherited macular dystrophy with an estimated prevalence between 1:8000 and 10,000 [3]. Due to phenotypic and genotypic variability, an accurate prevalence of Stargardt macular dystrophy cannot be estimated. From a clinical point of view, patients show progressive atrophy of the cellular layers in the central macula beginning from the RPE, fundus flecks derived from the accumulation of RPE lipofuscin in the retina, and peripapillary sparing (the tissue adjacent to the optic nerve is generally spared of disease changes). Unfortunately, the only sign that can be recognized in almost all patients is maculopathy: the disease always begins in the central macula. To further complicate a clinical diagnosis, the age at onset, disease progression, and family history can considerably vary among patients. In many cases, a family evaluation gives just a few pieces of data to support the diagnosis. In fact, the most frequent *ABCA4*-related Stargardt macular dystrophy shows an autosomal recessive model of transmission, and many affected people are sporadic cases. Conversely, when genetic evaluation reveals an autosomal dominant transmission pattern, a more specific gene panel can be evaluated in the first-tier analysis. In these cases, *PRPH2*, *ELOVL4*, and *PROM1* are the most frequent causes of macular dystrophy, but many other genes should be taken into consideration. In fact, although classical Stargardt disease (STGD) genes according to OMIM classification include *ABCA4*, *ELOVL4*, and *PROM1*, the massive groups of macular dystrophies (HP:0007754) and macular degenerations (HP:0000608) comprise more than 100 genes (according to human phenotype ontology (HPO)). An accurate genetic test for diagnostic purposes should evaluate all the known genes potentially associated with the phenotype. This kind of analysis can be performed through the application of whole-exome sequencing with phenotype-driven filtering. Unfortunately, even when this broad-range analysis is performed, some patients remain without genetic confirmation of their diagnosis. This may be due both to the limitations of whole-exome sequencing, which cannot genotype deep-intronic variants, and the limitation in the knowledge regarding the genes responsible for macular dystrophies. In this scenario, extensive whole-exome analysis together with an accurate application of filtering algorithms may help to define new genetic etiologies. In the presented family, an analysis of the known genes responsible for macular dystrophy did not reveal pathogenic mutations. The consanguineous pedigree together with an extensive evaluation of the whole-exome data allowed the identification of the splicing variation in the *RDH8* gene. Mutations in the *RDH8* gene have not been reported before in human phenotypes.

The *RDH8* gene encodes all-trans-retinol dehydrogenase (RDH8), a visual cycle enzyme that reduces all-trans-retinal in rod and cone outer segments [18,19,20]. The clearance of all-trans-retinal in photoreceptors is mediated by two fundamental players: ABCA4, which translocates all-trans-retinal from the inside to the outside of photoreceptor outer segment discs, and RDH8, which reduces all-trans-retinal in all-trans-retinol in the cytosolic lumen of POS [16,21]. When the function of one of these players is compromised, the excess of all-trans-retinal produces several bis-retinoids (i.e., A2E) that are resistant to hydrolysis. Bis-retinoids accumulate in lipofuscin granules, leading to RPE cell and photoreceptor death [22]. Murine and cellular models with the absent activity of RDH8 and/or ABCA4 showed similar pathological features, which ranged from age-related macular dystrophy (AMD) to Stargardt disease [16,17].

In the present work, a novel splicing variant (c.262+1G>A) in *RDH8* is described. The substitution of guanine in alanine immediately after the nucleotide 262 causes aberrant splicing with the retention of a second exon (Figure 5). The C-terminal segment (exon 6) of the *RDH8* gene demonstrates membrane-binding properties [23]. No functional studies have reported the function of other *RDH8* regions.

The identification of RDH8 as a member of the visual cycle is quite recent [18]. From a molecular point of view, RDH8 reduces all-trans-retinal to all-trans-retinol in the presence of NADPH [18]. It is located in the outer segments of photoreceptors, and it is also named photoreceptor retinol dehydrogenase (PRRDH) [21]. The human *RDH8* gene [NM_015725.4] encodes a 311-amino acid protein [ENST00000591589.3] and three other transcripts, of which there are two splice variants (ENST00000651512.1, ENST00000587782.1) and a non-coding transcript (ENST00000589570.1). *RDH8* has been largely studied in mice, in which it is known as prRDH. *RDH8* plays a crucial role in the reduction of all-trans-retinal in rod and cone outer segments [19,20].

The cascade of molecular steps involved in the visual process begins in photoreceptors with rhodopsin activation, which takes place through the conversion of chromophore 11-cis-retinal to all-trans-retinal. From this first step, a signaling cascade begins that ends with the visual stimulus to the brain. The all-trans-retinal in the photoreceptors is then cleared through a two-step mechanism: (i) all-trans-retinal is translocated from the inside to the outside of photoreceptor outer segment (POS) discs (mediated by ATP-binding cassette transporter 4-ABCA4) and (ii) the reduction of all-trans-retinal in all-trans-retinol in the cytosolic lumen of POS (mediated by retinol dehydrogenase 8-RDH8) [16,21]. Functional analyses of RDH8 activity showed that the Rdh8−/− outer segments of mouse rod cells can convert about 13% of their total chromophore to retinol; this activity is substantially lower than the activity observed in wild-type cells (about 80%) [21].

In Rd8−/− mice, the high levels of all-trans-retinal determine the accumulation of condensation products, such as A2E (di-retinoidpyridinium-ethanolamine) and all-trans-retinal dimer (RALdi). These condensation products of all-trans-retinal are lipofuscin fluorophores that generally accumulate with age in the RPE. The RPE is a layer of epithelial cells that are responsible for the phagocytic removal of several products. Retinal lipofuscin is a mixture of fluorescent cross-linked proteins and lipids that accumulate in the lysosomes of RPE cells. Among the lipid bis-retinoids found in retinal lipofuscin, A2E and RALdi are the most represented [22].

To maintain the efficient functioning of the molecular cascade, the visual cycle provides a re-conversion of all-trans-retinal in 11-cis-retinal [24,25]. The last phase of all-trans-retinal recycling is cone-specific because it involves the conversion (oxidation) of 11-cis-retinol into 11-cis-retinal. This process is fundamental for pigment regeneration and dark adaptation [17,26,27]. A functional study reported that the deletion of *RDH8* in cone mouse cells blocked their ability to oxidize 9-cis-retinol (a commercially available analogue of 11-cis-retinol), suggesting an indirect cone function of the RDH8 enzyme [17]. In fact, *RDH8/ABCA4*-deficient cones cannot oxidize 9-cis-retinol to regenerate visual pigment and recover their photosensitivity [17]. In particular, it is hypothesized that 11-cis-retinol is oxidized into 11-cis-retinal and then converted by lipid-catalyzed isomerization into all-trans-retinal. All-trans retinal levels are kept in check by RDH8, which reduces all-trans-retinal into all-trans-retinol.

The increase in all-trans-retinal levels led to the accumulation of bis-retinoids, such as A2E. [28,29]. Furthermore, an overload of all-trans-retinal levels induces photoreceptor degeneration through different mechanisms. At the basis of all mechanisms, the accumulation of bis-retinoids in RPE cells has been recognized. Through phagocytosis, RPE cells incorporate bis-retinoids in lysosomes. Typically, bis-retinoids are resistant to hydrolysis and accumulate in lipofuscin granules [30]. This progressive accumulation is physiologically driven by age or pathologically accelerated by genetic mutations (as seen in ABCA4 models). The retinal lipofuscin accumulation led to RPE cell death and the secondary death of photoreceptors [22]. One of the hypothesized pathways of photoreceptor cell death is ferroptosis, an iron-dependent form of nonapoptotic cell death. This mechanism is particularly interesting for therapeutic purposes. In fact, in murine models, intraperitoneal injection of the ferroptosis inhibitor Fer-1 reduced photoreceptor atrophy. Another pathological mechanism involves inflammatory pathways. In fact, endogenous products derived from an aberrant function of RDH8 and ABCA4 stimulate TLR3, leading to cellular apoptosis and retinal inflammation. This hypothesis was supported by the observation that mice Tlr3−/−Rdh8−/−Abca4−/− are protected from cone–rod dystrophy (CORD) if compared with Rdh8−/−Abca4−/− mice [31]. Interestingly, the pathologic cascade seems to be driven by light exposure, and it is mediated by inflammation. Room light exposure led to chronic inflammation and retinal degeneration; intense light exposure led to a transient increase in inflammatory molecules, which return to the basal level 7 days after light exposure [32].

In the present family, the filtering algorithm applied to the whole-exome data took into consideration the pedigree (consanguinity), phenotype, and gene function. Unfortunately, many cases are sporadic cases and family data are missing. During the last century there has been an improvement in sequencing technologies, and, presently, the rapid improvement in artificial intelligence (AI) can support the development of tools and algorithms for medical and genetic purposes. Many multifactorial disorders yet benefit from the computational evaluation of individualized risk (i.e., age-related macular dystrophy) [33]. The application of AI to Mendelian disease is more difficult due to the frequent absence of family data. Nevertheless, some interesting applications of AI have been developed in acutely ill infants, in which trios were available [34]. Although the medical contribution remains an indispensable requisite for the correct interpretation of the results derived from the AI protocols [35], it is expected that in the future, the improvement of AI protocols will allow for the correct genetic diagnosis in many sporadic cases.

In the presented family, the application of whole-exome sequencing and validation through a multilevel approach (international database searching, prediction scores calculation, splicing analysis assay, segregation analyses) allowed the identification of a splicing mutation in a gene that has never been reported before in human disorders. The ophthalmological phenotype described in this family supported what was previously reported in experimental models of *RDH8*. This case further demonstrates the importance of genetic counselling to provide an accurate family history collection. Without evidence of consanguinity and a family history of the disorder, the prioritization of the exome variant would not allow for the selection of the RDH8 causative variant. To our knowledge, this is the first family in which a deleterious mutation in *RDH8* was associated with a human phenotype. Almost confirming what has been previously reported in experimental models of disease [16,17], the family described herein showed a Stargardt disease phenotype. From an ophthalmological point of view, it resembled a fall into the 10% of patients with a diagnosis of Stargardt disease that lack mutations in *ABCA4* or genes associated with similar phenotypes. Nevertheless, the application of an individualized filtering algorithm on WES data permits the identification of the responsible mutation in the *RDH8* gene. In this family, a family and phenotype-driven algorithm together with NGS technology allows the identification of a homozygous splicing mutation in *RDH8*, confirming the genetic etiology of the phenotype. Further studies are needed to clarify how this mutation modifies the RDH8 ensemble on the membrane and in the cell.

## 5. Conclusions

In conclusion, an *RDH8* splicing mutation in a family exhibiting an ophthalmological Stargardt-like phenotype has been identified through a whole-exome sequencing approach. This is consistent with previous studies on experimental animals and cells in which lack of *RDH8* impaired photoreceptor function. Due to molecular function, it is hypothesized that experimental therapeutical approaches developed for *ABCA4*-maculopathy could be also applied to *RDH8* patients. Furthermore, the application of next-generation sequencing in patients with a Stargardt-like phenotype can support the rapid identification of mutations in *ABCA4* and other genes associated with similar phenotypes, also allowing the detection of new involved genes, such as *RDH8*.

## Figures and Tables

**Figure 1 genes-14-01659-f001:**
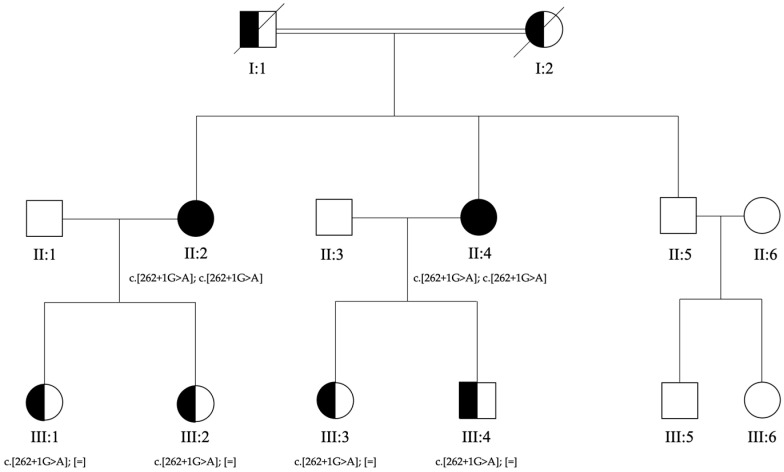
Pedigree of the family. Genotypes for c.262+1G>A in *RDH8* gene are reported below each tested individual. The affected sisters (II:2 and II:4) received a clinical diagnosis of Stargardt macular dystrophy. Their parents (I:1 and I:2) were reported as sighted but never evaluated. The healthy brother (II:5), son (III:4), and daughters (III:1, III:2, and III:3) of the affected sisters were evaluated by ophthalmologists who confirmed the absence of signs of macular disease.

**Figure 2 genes-14-01659-f002:**
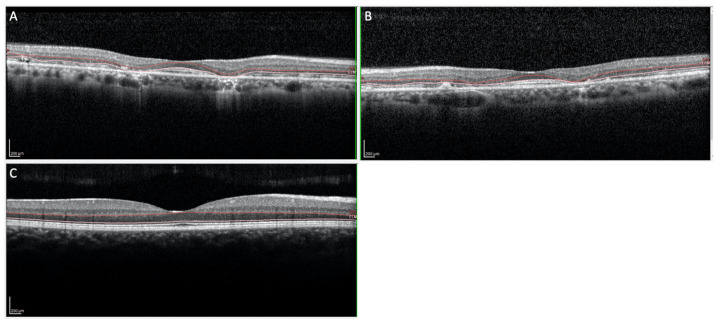
(**A**,**B**) OCT imaging of the macular region showing subretinal deposits in the foveal region and thinning of the outer nuclear layers ((**A**): Right eye; (**B**): Left eye). (**C**) OCT imaging of the normal macular region.

**Figure 3 genes-14-01659-f003:**
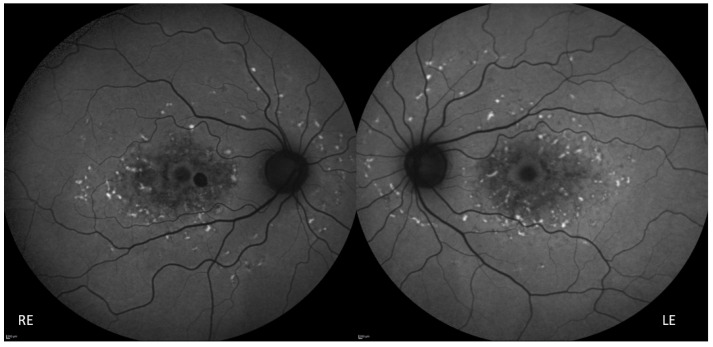
Fundus autofluorescence showing perifoveal and peripheral flecks as well retinal pigment epithelium atrophy in the foveal and perifoveal region. RE: Right Eye, LE: Left Eye.

**Figure 4 genes-14-01659-f004:**
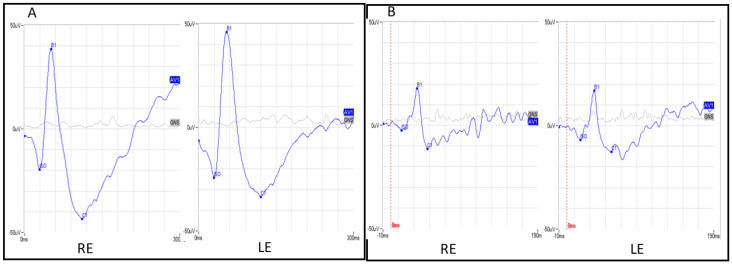
(**A**) Ganzfeld mixed rod–cone electroretinograms in both eyes. (**B**) Ganzfeld cone electroretinograms in both eyes. RE: Right eye, LE: left eye.

**Figure 5 genes-14-01659-f005:**
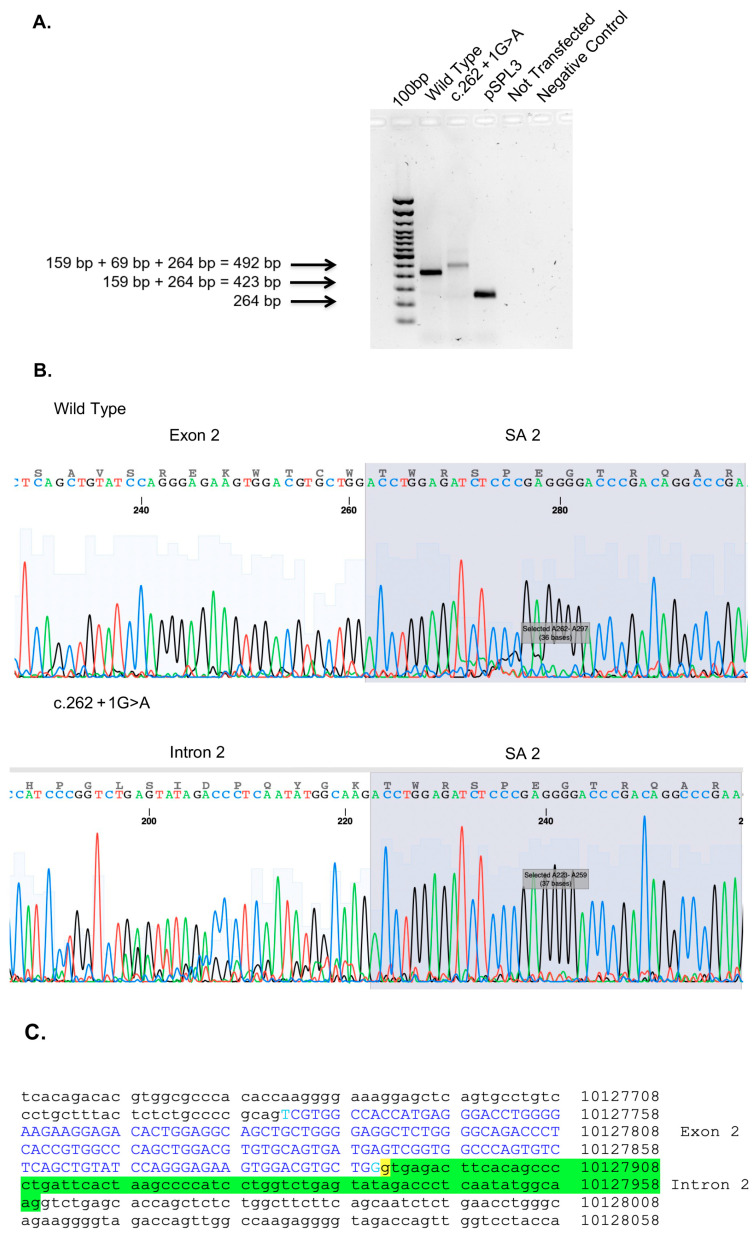
Minigene assay, Sanger sequencing of RHD8. (**A**) Agarose gel shows RT-PCR results of minigene assay for variant c.262+1G>A. Lane 1 shows an amplicon of 423 bp corresponding to a wild-type genotype (159 bp of normal splicing of exons 2 + 264 bp of pSPL3 exon); lane 2 shows an amplicon of 492 bp corresponding to abnormal splicing produced by mutation c.262+1G>A (492 bp [159 bp of normal splicing (exon 2) + 69 bp of partial retention of intron 2] + 264 bp of pSPL3 exon); lane 3 shows the amplification of pSPL3 without RHD8 cloning; lane 4 shows the amplification of HEK 293 T cDNA without transfection of pSPL3; and lane 5 shows negative control of PCR amplification. (**B**) Sanger sequence shows the normal sequence and the partial retention of intron 2. (**C**) Genomic sequence of RHD8 exon 2 is in blue capital letters, underlined in yellow is c.262+1G>A variant, and underlined in green is the partial retention of intron 2.

## Data Availability

Data available on request due to privacy restrictions.

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
