# Peer review of "A Splicing Variant in RDH8 Is Associated with Autosomal Recessive Stargardt Macular Dystrophy"

_genes, 2023, doi:10.3390/genes14081659_

Round 1

Reviewer 1 Report

In the study, Zampatti et al found a splicing variant in RDH8 associated with autosomal recessive Stargardt macular dystrophy. The molecular bases of Stargardt macular dystrophy are now mainly known to be caused by ABCA-related mutations. However, it has been reported that mutations in several other genes could also cause Stargardt-like macular dystrophies. In this study, Zampatti et al newly identified that the mutation in RDH8 is associated with Stargardt-like macular dystrophies. These results are highly suggestive not only for understanding Stargardt-like macular dystrophy but also for gaining insights into the functions of RDH8 in the retina. However, as described below, there are several points that need improvement to make the manuscript more understandable. In my opinion as a reviewer, this paper requires revision.

Point 1: Line 68, "eep-intronic variants" should be "deep-intronic variants."

Point 2: Presentation in Fig 2. The intended audience for this study includes not only ophthalmologists but also general readers. Therefore, notations like "OD" and "OS" are not appropriate. Additionally, general readers may not grasp the information presented in Figure 2B. It would be helpful to include some annotations in the figures and increase the size of the scale bars and attached numbers to improve visibility. Please address these points.

Point 3: Line 150, "thinning of the outer nuclear layer." Please include the control image and clearly indicate the locations of the outer nuclear layer in the panels. It would be preferable to compare the results with normal human data.

Point 4: Line 161, "Ganztels rod-cone and cone electroretinograms were of normal amplitude in both eyes." Do you mean "Ganzfeld rod-cone and cone electroretinograms"? The ERG results seem very important, so please provide these data.

Point 5: Line 207, "the retention results in an insertion of 9 amino acids (GETSQPLIH) and a premature stop codon (Figure3)." In Figure 3, there are no indications corresponding to the statement in line 207.

Point 6: Figure 3. What are NT and CN in the PCR gel image? What are Esone2 and IVS2 in Sequenza? Is Sequenza an Italian term?

The localization of the caption "c.262+1G>A" in the Sequenza panel is too close to the upper WT results, and it is confusing.

Author Response

Dear Reviewer 1, 

thank you very much for your comments which have really improved the manuscript.

Please, find our point-by-point response:

Point 1: Line 68, "eep-intronic variants" should be "deep-intronic variants."

Response 1: the typo has been corrected

Point 2: Presentation in Fig 2. The intended audience for this study includes not only ophthalmologists but also general readers. Therefore, notations like "OD" and "OS" are not appropriate. Additionally, general readers may not grasp the information presented in Figure 2B. It would be helpful to include some annotations in the figures and increase the size of the scale bars and attached numbers to improve visibility. Please address these points.

Response 2: We included annotations explaining the Figure 2, 3 and 4. We also increased the scale bar in Figure 2 and, in the same figure, included a normal retinal profile at OCT, for comparison

Point 3: Line 150, "thinning of the outer nuclear layer." Please include the control image and clearly indicate the locations of the outer nuclear layer in the panels. It would be preferable to compare the results with normal human data.

Response 3: See point 2 above.

Point 4: Line 161, "Ganztels rod-cone and cone electroretinograms were of normal amplitude in both eyes." Do you mean "Ganzfeld rod-cone and cone electroretinograms"? The ERG results seem very important, so please provide these data.

Response 4: The word Ganzfeld has been corrected as appropriate. Ganzfeld ERG results are reported in the new Figure 4.

Point 5: Line 207, "the retention results in an insertion of 9 amino acids (GETSQPLIH) and a premature stop codon (Figure3)." In Figure 3, there are no indications corresponding to the statement in line 207.

Response 5: The figure 3 and its caption have been corrected. Figure 3 has been split in three parts: agarose gel showing RT-PCR results of minigene assay (A), sanger sequence comparing wild-type and mutated sequence (B), genomic sequence highlighting retained region.

Point 6: Figure 3. What are NT and CN in the PCR gel image? What are Esone2 and IVS2 in Sequenza? Is Sequenza an Italian term?

The localization of the caption "c.262+1G>A" in the Sequenza panel is too close to the upper WT results, and it is confusing.

Response 6: Figure 3 has been fully revised. Agarose gel image have been corrected (A). Genomic sequence of exon 2 and part of intron 2 of RDH8 gene were included in the figure (C) to guide the evaluation of sanger sequencing results (B)

Reviewer 2 Report

Zampatti et al., identified a novel homozygous splice variant in the RDH8 causing Stargardt macular dystrophy. Overall, the manuscript is concise and well-written. Please made some necessary changes.

1.     In line 89, correct fig.1 with Figure 1.

2.     Elaborate on the legends of Figure 1, and describe what phenotypes these individuals show (I:1, I:2, III:1, III:2, III:3, III:4). please write the genotypes below each individual which were sequenced for RDH8 splice variant.

3.     Please mention the individuals who donated the blood for genomic DNA analysis (line 95, 96).

4.     What was the overall coverage of exome data, mention in section 2.2? What cut-off value was used to prioritize the variant based on allelic frequency?

5.     In addition to RDH8, after prioritization how many variants were short-listed, please add the annotated information of each variant/gene and associated phenotypes. (Add this information in a supplementary table)

6.     Did the author check compound. het variants in the exome data?   

7.     Please check the allelic frequency of the variant in “BRAVO” and “all of us” databases and mention it in the text (line 128, 129).

8.     The author mentioned that “Sanger sequencing was performed on all affected patients (II:2 and II:4) and in their sons (III:1, III:2, III:3, and III:4) to confirm expected genotypes” while individuals III:1, III:2 and III:3 are shown as females in the pedigree (Figure 1) please correct it (Line 181).

9.     Did the author submit the variant information in clinvar? If yes, please add the accession number in the text.

10.  Please add the NM number and genomic coordinate of the variant (line 177).

11.  Please the CADD score (line 198).

12.  What is the ADA and RF score of the splice variant? mention it in the results section of the manuscript.

13.  Is there any limitation of the study? If yes, please mention it in the discussion.

Author Response

Dear Reviewer 2, 

thank you very much for your comments which have really improved the manuscript.

Please, find our point-by-point response:

Point 1.     In line 89, correct fig.1 with Figure 1.

Response 1: the typo has been corrected

Point 2.     Elaborate on the legends of Figure 1, and describe what phenotypes these individuals show (I:1, I:2, III:1, III:2, III:3, III:4). please write the genotypes below each individual which were sequenced for RDH8 splice variant.

Response 2: Legends of Figure 1 has been implemented with phenotypes of evaluated individuals. Figure 1 has been completed with genotype data of each tested individual.

Point 3.     Please mention the individuals who donated the blood for genomic DNA analysis (line 95, 96).

Response 3: The paragraph has been implemented with a sentence that describes type of samples evaluated for each individual.

Point 4.     What was the overall coverage of exome data, mention in section 2.2? What cut-off value was used to prioritize the variant based on allelic frequency?

Response 4: Data from exome assay has been implemented in section 2.2. The cut-off value used to prioritize variants were allele frequency < 0.05 (according to BA1 criterion from ACMG 2015). The cut-off has been included in 3.2 section (line 179).

Point 5.     In addition to RDH8, after prioritization how many variants were short-listed, please add the annotated information of each variant/gene and associated phenotypes. (Add this information in a supplementary table)

Response 5: In section 3.2 data about number of selected variants after prioritization has been added. The complete table with exome variants cannot be uploaded because it can identify the sample, that should remain anonymous. Data about number and interpretation of exome variants has been added in the manuscript (lines 186-203).

Point 6.     Did the author check compound. het variants in the exome data?   

Response 6: After prioritization, only four variants have been reported as expressed in retina. Three of these variants were heterozygous in different genes (RRH, GPSM1, and NEDD8-MDP1). No potentially heterozygous compound variants were found in the selected genes. One of these variants was homozygous in RDH8 gene.

Point 7.     Please check the allelic frequency of the variant in “BRAVO” and “all of us” databases and mention it in the text (line 128, 129).

Response 7: Allelic frequency of the variant was also checked in BRAVO and All of Us databases. Methods and results have been implemented with these data (sections 2.4 and 3.3)

Point 8.     The author mentioned that “Sanger sequencing was performed on all affected patients (II:2 and II:4) and in their sons (III:1, III:2, III:3, and III:4) to confirm expected genotypes” while individuals III:1, III:2 and III:3 are shown as females in the pedigree (Figure 1) please correct it (Line 181).

Response 8: The sentence has been corrected

Point 9.     Did the author submit the variant information in clinvar? If yes, please add the accession number in the text.

Response 9: Yes, the variant has been submitted in ClinVar. We are waiting for approval, submission number has been added to the manuscript.

Point 10.  Please add the NM number and genomic coordinate of the variant (line 177).

Response 10: NM number and genomic coordinate has added to the sentence

Point 11.  Please the CADD score (line 198).

Response 11: CADD score has added to the sentence.

Point 12.  What is the ADA and RF score of the splice variant? mention it in the results section of the manuscript.

Response 12: Data about ADA and all prediction scores listed were added to the sentence

Point 13.  Is there any limitation of the study? If yes, please mention it in the discussion.

Response 13: There are no limitation of the study.

Round 2

Reviewer 1 Report

In the revised manuscript, Zampatti et al revealed the data more precisely, but the interpretations of the data originally revealed in line 148 to line 162 in previous version was omitted in the revised version. The explanation and interpretation of the data is important. Therefore, please retain the explanations.

Author Response

Dear reviewer, 

thank you very much for your suggestion. Actually, in the revised version of the manuscript, two explanation of ophthalmological  evaluations were accidentally deleted. We provided the reintroduction of these sentences (lines 156-161). 

Thanks for your support